# 'Who Cares?' The experiences of caregivers of adults living with heart failure, chronic obstructive pulmonary disease and coronary artery disease: a mixed methods systematic review

Miriam Catherine Noonan,[1] Jennifer Wingham,[2] Rod S Taylor[3]

[1]European Centre for Environment and Human Health, University of Exeter Medical School, Knowledge Spa, Royal Cornwall Hospital, Truro, UK
[2]Royal Cornwall Hospitals NHS Trust, Research, Development and Innovation, F37, Knowledge Spa, Royal Cornwall Hospital, Truro, Cornwall & University of Exeter, Exeter, UK
[3]University of Exeter Medical School, Exeter, UK

**Correspondence to**
Miriam Catherine Noonan;
mn354@exeter.ac.uk

## ABSTRACT

**Objective** To assess the experiences of unpaid caregivers providing care to people with heart failure (HF) or chronic obstructive pulmonary disease (COPD) or coronary artery disease (CAD). **Design** Mixed methods systematic review including qualitative and quantitative studies. **Data sources** Databases searched: Medline Ebsco, PsycInfo, CINAHL Plus with Full Text, Embase, Web of Science, Ethos: The British Library and ProQuest. Grey literature identified using: Global Dissertations and Theses and Applied Sciences Index and hand searches and citation checking of included references. Search time frame: 1 January 1990 to 30 August 2017.

**Eligibility criteria for selecting studies** Inclusion was limited to English language studies in unpaid adult caregivers (>18 years), providing care for patients with HF, COPD or CAD. Studies that considered caregivers for any other diagnoses and studies undertaken in low-income and middle-income countries were excluded. Quality assessment of included studies was conducted by two authors.

**Data analysis/synthesis** A results-based convergent synthesis was conducted.

**Results** Searches returned 8026 titles and abstracts. 54 studies—21 qualitative, 32 quantitative and 1 mixed method were included. This totalled 26 453 caregivers who were primarily female (63%), with median age of 62 years. Narrative synthesis yielded six concepts related to caregiver experience: (1) mental health, (2) caregiver role, (3) lifestyle change, (4) support for caregivers, (5) knowledge and (6) relationships. There was a discordance between paradigms regarding emerging concepts. Four concepts emerged from qualitative papers which were not present in quantitative papers: (1) expert by experience, (2) vigilance, (3) shared care and (4) time.

**Conclusion** Caregiving is life altering and complex with significant health implications. Health professionals should support caregivers who in turn can facilitate the recipient to manage their long-term condition. Further longitudinal research exploring the evolution of caregiver experiences over time of patients with chronic cardiopulmonary conditions is required.

**Trial registration number** CRD42016053412

### Strengths and limitations of this study

► This mixed methods systematic review provides the opportunity for a broadened and deeper understanding of the qualitative and quantitative literature on the experiences of unpaid caregivers' providing care to people with heart failure, chronic obstructive pulmonary disease and coronary artery disease.
► This review provides an integration of the type and extent of caregiver's experiences and predictors of caregiver's experiences.
► To maximise applicability we included studies from higher income countries only.
► Quality of evidence limited by assessment of caregiver experience at single point of time and there is need for future studies that employ longitudinal or repeated measures design.

### INTRODUCTION

A caregiver is anyone providing unpaid care, to a friend or family member who is unable to care for themselves.[1] This may be emotional support; someone to talk to, or practical support; dressing wounds, mobility assistance or medication checking.[2] There are 43.5 million caregivers in the USA, 2.86 million in Australia and 6.5 million in the UK.[3] Between 2001 and 2011, the number of unpaid caregivers in the UK grew at a faster rate than population growth.[4] The annual value of unpaid care provided to an individual with a chronic illness is estimated to be £132 billion.[5]

Focus groups examining a caregivers' life conducted by 'The Institute of Public Care' (2017), based at Oxford Brookes University; described caregivers as the 'Skilled Helper' performing a series of roles.[6] Seltzer and Li describe a dynamic process of transitions to being a caregiver.[7] These transitions comprise participating in the role before identifying as

a caregiver, acceptance of the role, engaging in it with awareness and sometimes moving beyond the caregiving role when the patient moving to paid care settings or bereavement occurs. This process is not linear and people move through the different transitions at varying rates. Acknowledging this, it is imperative for caregivers to receive a caregiver needs assessment as legally stipulated by the 2014 Care Act.[8] Additionally, the National Institute for Health and Care Excellence clinical guidelines for heart failure (HF) (CG108)[9] and chronic obstructive pulmonary disease (COPD) (CG101)[10] both recommend that family members or caregivers are provided with support and included in discussions about care.

Cardiopulmonary disease is a primary cause of illness. Cardiovascular disease is responsible for 45% deaths in Europe[11] and one in four deaths in the USA.[12] By 2020, COPD is projected to be in the global top five of diagnoses causing years lost through early mortality or disability-adjusted life years.[13] Caregivers of patients with HF have a multitude of unmet needs due to fluctuations in the trajectory of HF.[14] COPD has frequent unplanned hospital admissions and a high morbidity rate.[15] Caregivers experience depressed mood, greater anxiety and increased subjective burden when their support needs are not met.[16 17] The unpredictability of HF and COPD leads to caregivers constantly adjusting their role, creating a need to continuously reassess what caregiver needs are.[18 19] Spousal caregivers of patients with myocardial infarction experience increased levels of stress, lifestyle impact and emotional distress.[20] Caring for coronary artery bypass graft patients in tasks such as monitoring and provision of emotional support increased caregiver burden to a level described as moderate.[21] COPD and cardiovascular disease are both increasing in prevalence and frequently coexist.[22 23] We know of no systematic review that synthesises quantitative and qualitative studies to combine caregivers' experiences of people with HF, COPD or coronary artery disease (CAD).

Using a mixed methods systematic review methodology including both qualitative and quantitative literature, this study aims to understand the experiences of adult caregivers when supporting people with HF, COPD or CAD.

## METHODS
We conducted and reported this systematic review in accordance with the Preferred Reporting Items for Systematic Reviews and Meta-Analyses (PRISMA) statement.[24]

### Patient and public involvement
There was no patient and/or public involvement in this systematic review.

### Study design
This study employed a mixed methods systematic review assessing both qualitative and quantitative studies.[25] The rationale for using a mixed methods review approach was multifaceted. First, to gain a qualitative assessment of

the type and extent of caregiver's experiences. Second, to assess the quantitative predictors of caregiver's experiences. Third, to develop a holistic perspective of what caregiver experiences. Finally, we wanted to assess the degree of convergence between qualitative and quantitative experiences.

### Search strategy
Our search strategy was designed in conjunction with a Health Services Librarian and Information Specialists. Search terms included condition-specific terms, that is, 'heart failure', 'COPD' and 'coronary artery disease', caregiver-specific, plus experience related terms, 'experience', 'quality of life' 'activities of daily living', 'occupational engagement', 'time use', 'self-efficacy', 'coping strategies', 'leisure activity', 'information exchange' and 'caregiver expectation' (see online supplementary file 1, table 1 for complete list of search terms). Databases searched included: Medline Ebsco, PsycInfo, CINAHL Plus with Full Text, Embase, Web of Science, Ethos: The British Library and ProQuest. Grey literature was identified using Global Dissertations and Theses and Applied Sciences Index and hand searches and citation checking of included references. To ensure the contemporary nature of the evidence considered, the search time frame was January 1990 to August 2017. A single researcher (MN) initially screened titles and abstracts. Selection of full papers was performed by two researchers (MN and either JW or RST) and cross-checked with the eligibility criteria.

### Study selection
Studies were included if they addressed 'caregiver experience', which was defined as encompassing the daily activities of caregivers and the impact of these activities on their lives. These were English language studies involving unpaid adult caregivers (aged >18 years), providing care for patients with HF, COPD or CAD living in the community and not residential settings with paid care staff. Qualitative, quantitative and grey literature studies were all included in the search strategy. Conference papers were excluded. Outcomes of interest included psychological and physical outcomes reported, occupational engagement and routine. As we sought to inform the practice of the UK and other high-income countries, we excluded studies undertaken in low-income and middle-income countries.[26]

### Data extraction
Data extracted from retained studies included: study design, sample and recruitment, study description, method, findings, discussion and authors' conclusions and limitations. Caregiver quotes were extracted from qualitative studies. For quantitative studies, data extraction also included details of attrition and data analysis.

### Study quality assessment
Qualitative studies were appraised using the Critical Appraisal Tool.[27] In absence of an existing quality tool

that could be used to appraise quantitative studies addressing the specific question of this study, a quality assessment tool was developed by the research team based on what were deemed to be the appropriate core biases, that is, (1) was the study design longitudinal (score of 1) or cross-sectional (score of 0); (2) how was the sample recruited? Purposive (score of 1) or convenience (score of 0); (3) was the level of attrition/response rate acceptable? Attrition of 20%/lower or response rate of 80% or above (score of 1) or attrition of >20% or response rate <80% (score of 0); (4) was a validated quantitative outcome(s) used? Validated (score of 1), non-validated (score of 0); (5) were the methods of data analysis appropriate? Multivariate (score of 1) or univariate (score of 0). Based on their quality assessment, scores were totalled and studies were ranked: 1 or 2 'low quality', 3 'medium quality' and 4 or 5 'high quality'. Data extraction and quality appraisal was first conducted by a single researcher (MN) and checked by one of two researchers (JW or RST).

## Data analysis and synthesis

The methodology of mixed methods data synthesis is an emerging one and no single approach has yet been universally accepted.[28] In this study, a results-based convergent design was chosen.[29 30] This requires transformation of one method into another. Due to the heterogeneity of the quantitative methods, a meta-analysis was not appropriate. Instead, applying a narrative profile formation, quantitative data were converted into qualitative data.[31] Extracted data from quantitative and qualitative studies were imported into separate spreadsheets. A meta-ethnographic approach was used to synthesise qualitative studies.[32] A narrative formation approach[33] was used to synthesise the quantitative data into a qualitative data set. Narrative formation is a verbal description via the use of profiles of each of the studies. The five profiles are modal, average, holistic, comparative and normative.[33] Table 1 provides an example of this approach. This resulted in two qualitative data sets[34] from which concepts emerged. A mapping table was completed in order to provide an audit trail of how the overall concepts across all papers were derived (see online supplementary file 1, table 2a, b and c). Initial synthesis was conducted by a single researcher (MN) and corroborated by two experienced researchers in quantitative (RST) and qualitative (JW) research.

## RESULTS
## Study selection

Study selection process is summarised in a PRISMA flow diagram shown in figure 1. Following removal of duplicates, the search strategy yielded a total of 8026 titles and abstracts. Of these, 242 full papers were reviewed, of which 57 papers (54 studies) were included for synthesis. A detailed summary of included studies is provided in table 2. A comprehensive outline of study results and concepts generated by each study is included in online supplementary file 2).

## Study characteristics

Of the 54 studies, 21 were qualitative, 32 quantitative and 1 mixed methods. Thirty-four focused on HF, 14 COPD and 6 CAD. The total number of caregiver participants was 26 453. Caregivers were primarily female (63%), with a median age of 62 years. Patient median age was 69 years. A summary of study characteristics is provided in table 3.

## Quality assessment

Studies of insufficient quality were excluded, qualitative papers were appraised and only high-quality qualitative studies were included.[35] A total of 21 out of 193 qualitative studies were classified as high quality. Quantitative studies were classified as follows: 3 high quality, 12 medium quality and 17 low quality (see table 4(a) and (b) for quality appraisal). Given the number of high-quality qualitative studies and in accord with current guidelines for the synthesis of qualitative evidence, we limited inclusion to qualitative studies of high quality only.[35] In contrast, given the low number of high-quality quantitative studies, to ensure comprehensiveness of our review, we included all quantitative studies, regardless of quality.

## Findings

Six concepts relating to caregiver experience were identified: (1) mental health, (2) caregiver role, (3) lifestyle change, (4) support for caregivers, (5) knowledge and (6) relationships. Four additional concepts were identified from qualitative papers only (6) expert by experience, (7) vigilance, (8) time and (9) shared-care (figure 2). The concepts are reflected in caregiver quotes in table 5.

## Mental health

Twenty-five quantitative,[36–62] 20 qualitative[63–83] and 1 mixed methods[84] study addressed mental health. This encompassed depression and burden. Caregivers described an internal and external conflict of emotions, recognising a psychological change within themselves and the care recipient. Maintaining hope and positivity, versus managing worries, fears and anxieties was predominant.[62–82] The HF study by Pressler et al identified caregivers had moderately poor health at baseline and at 8 months but they had fewer depressive symptoms over time.[54] Burden arose due to greater responsibilities.[65 68 73 81 82] Yeh and Bull noted the quality of relationship and lack of family support significantly predicted greater family caregiver burden.[53] Näsström et al reported caregiver burden was concerned with the future and their fears of potential demands.[84] Those with greater resiliency appeared to adjust and cope better with the illness trajectory.[64 65 76 77] Caregivers described mental adjustment after an acute event.[77] Living through an acute event was long lasting and some experienced post-traumatic symptoms.[80]

**Table 1** Narrative formation (Chung, 2016)

| Profile | Explanation of profile | Modal narrative extracted from study | Emerging from study | Concept |
|---|---|---|---|---|
| Modal | This is a narrative description of the group being studied | 102 dyads, predominantly spousal caregivers. Group is caregivers of patients with HF. Comparing depressed and non-depressed patients with HF. | This study considers whether caregiver experiences are different for caregivers when caring for depressed or non-depressed patients. Spouse as a caregiver. Impact of depression on the caregiver. | |
| Average | This is a detailed narrative description based on the mean (average) attributes of the individuals/situations being studied | Caregiver mean age 56.7, 78% female, white (94%). 41% of patients New York Heart Association III-IV. '42% caregivers reported severe burden (the Zarit Burden Interview (ZBI)≥17)'.[36] | Caregiving resulted in burden experienced by caregivers in this study. | Mental health (burden) |
| Comparative | This is a description comparing studies and comparing individuals being studied | Caregivers who provided care for depressed patients reported higher burden than those caring for non-depressed patients. Caregivers related their burden to social life limitations, poor perceived control, stress of family obligations and patients' dependency. 'Caregivers of patients with depressive symptoms had a higher level of burden (25±13 vs 13.5±12 on the ZBI; p<0.001), spent more time caregiving (37±12 vs 30±11 on the Oberst Caregiving Burden Scale; p=0.004) and reported worse mental quality of life (46±10 vs 51±10 on the SF-12v2; p=0.026) than those of patients without depressive symptoms'.[36] | Patient illness severity impacts on caregiver. Life changes negatively impacted on caregivers. Depressive symptoms of patients are associated with poor outcomes of caregivers. Caregiver's subjective and objective response to the patient's illness severity. | Lifestyle adjustment Mental health |
| Normative | A comparison of the study individuals with the general population | 27% of patients in this study scored 14 or higher on Becks Depression Index (a score of 14 or higher is clinically significant for depression). Family members caring for patients with HF with depressive symptoms had significantly higher levels of caregiving burden and worse quality of life compared with those caring for patients without depressive symptoms. 'Most difficult task for both sets of caregivers—providing emotional support (M=3.3, SD=1.2)'.[36] | Greater impact on caregivers' lives when patients were depressed. | Mental health Role of caregivers |
| Holistic (also called inferential or summative) | A description of the overall perception of the investigator | | Female, white caregivers, experienced greater levels of burden, loss of roles and greater distress when patients were depressed. This could be due to the increased need for practical and emotional support, feeling they need to be constantly present. | |

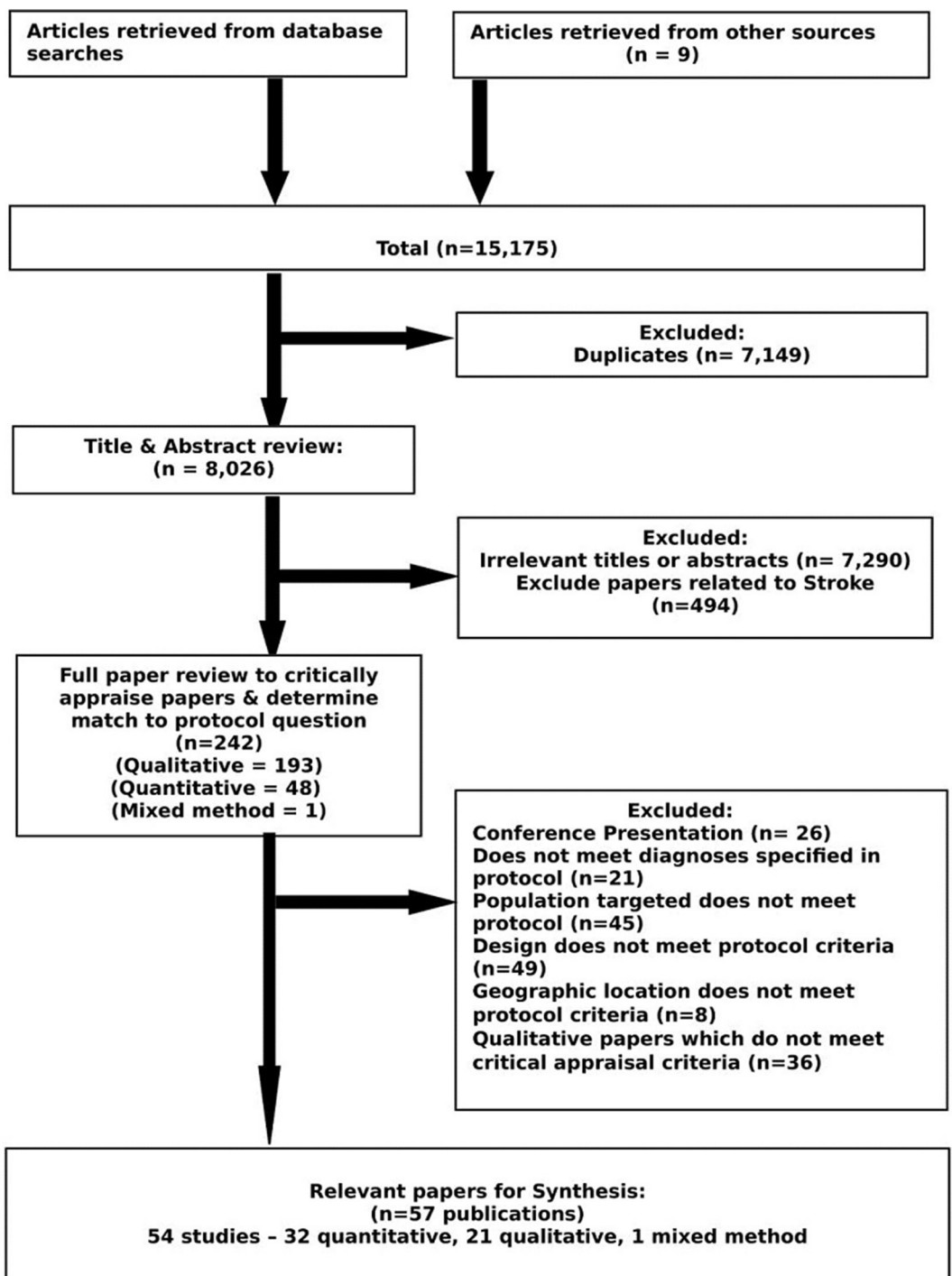

**Figure 1** Preferred Reporting Items for Systematic Reviews and Meta-Analyses flow diagram.

## Caregiver role

This is addressed in 18 qualitative[64–72 74 75 77–83 85] and 14 quantitative studies.[36 39–41 49–51 54 57 59 60 86–89] Caregiver role is complex and requires much coordination.[74 81 83] Caregivers describe significant role change such as increasing domestic tasks.[63 66 69 71 76–79 82] Role loss is prevalent[64 65 70] and caregivers need to reframe their identity.[72 80] Societal expectation regarding the relationship and gender, influences caregivers adjusting to their roles.[65 67 68 74 79] Caring can be positive and rewarding. Caregivers learn about themselves and strengths they have.[65 75–77 80 81 83] Pressler

*et al* described the tasks involved: domestic, emotional support, managing dietary needs and transport.[54] Pressler *et al* also reported that caregivers of persons with greater HF symptoms experienced more difficulty with their role.[54]

## Lifestyle changes

Fourteen quantitative[36 38–40 43 45 52 54 59 60 62 87 88 90] and 21 qualitative[63–83 85] studies addressed lifestyle changes. Caregivers experienced leisure, social and work-related problems.[36 39 90] Caring interrupted and eliminated

**Table 2** Summary of included studies

| First author/ref. no. | Diagnosis | Aims (as stated by authors) | Methods | Country | Data collection sampling | Participants caregiver (time caring) | Mean age (caregiver/recipient) |
|---|---|---|---|---|---|---|---|
| Ågren et al[37] | HF (NYHA II–IV) | Describe the levels and identify independent predictors of cg burden in partners of pts with HF. | QUANT (correlational) | Sweden | Cross-sectional purposive | 135 (101 F, 34 M) (NS) | 69/71 |
| Al-Rawashdeh[38] | HF (NYHA I–IV) | To examine whether individuals' disturbance predicted their own and their partners' QoL in HF | QUANT | USA | Cross-sectional purposive | 78 dyads (58 F, 20 M) (NS) | 62.2/59.5 |
| Andersen[63] | HF | Obtain knowledge on experiences and views and the desire for knowledge of family cgs of pts with HF, their competence and support required | QUAL | Norway | Interviews convenience | 19 (17 F, 2 M) (NS) | 63/NS |
| Badr et al[39] | COPD | Individual-level predictors of pt and and cgs depression in COPD as well as how dyad members effect each other's depression | QUANT | USA | Cohort study purposive | 89 (68 F, 21 M) (NS) | 54.8/67 |
| Bakas et al[40] | HF (NYHA II–IV) | Examine relationships among age, perceived control over managing HF perceived difficulty with tasks, perceived outcomes and perceived mental and general health among cgs of persons with HF | QUANT (descriptive correlational) | USA | Cross-sectional convenience | 21 (20 F, 1 M) (NS) | 59.6/62.7 |
| Baker et al[64] | HF (LVAD in situ) | To describe experiences of cgs of pts who received LVAD therapy as a bridge to transplantation | QUAL (phenomenological) (descriptive) | USA | Interviews convenience | 6 (5 F, 1 M) (26–372 days) | 51/NS |
| Bove et al[65] | COPD (GOLD C&D) | Explore how spouses of pts with severe COPD experience their role | QUAL | Denmark | Focus groups purposive | 22 (13 F, 9 M) (NS) | 69.4/NS |
| Burke et al[66] | HF (NYHA II–IV) | Understand what roles cgs perceive and desire for themselves, and to compare and contrast these roles with those they perceive to be desired by the healthcare system | QUAL (inductive) | USA | Interviews purposive | 20 (18 F, 1 M, 1NS) (<1to >8hours/week) | 59/64 |
| Chung et al[36] | HF (NYHA II–IV) | Examine differences in cg outcomes between cgs who care for pts with HF with and without depressive symptoms | QUANT | USA | Cross-sectional convenience | 102 dyads (79 F, 23 M) (NS) | 56.7/61.4 |
| Clark et al[85] | HF (NYHA II–IV) | To examine the complexity of caregiving for pts with HF | QUAL | Scotland | Interviews convenience | 30 (23 F, 7 M) (NS) | 68 F, 67 M/NS |
| Cossette[41] | COPD (GOLD III–V) | Examine relationship between type, number and disturbance of caring tasks and impact on mental health of cgs. Examine influence of social support | QUANT | Canada | Cross-sectional convenience | 89 (F) (mean=13years) | 65/68.6 |
| Evangelista et al[42] | HF (NYHA I–IV) | Describe emotional well-being of (descriptive) pts with HF and cgs (correlational) Identify factors associated with emotional well-being of pts and cgs. Determine gender differences in emotional well-being of pts and cgs | QUANT (descriptive) (correlational) | USA | Cross-sectional convenience | 103 dyads (73 F, 30 M) (NS) | 59.4/57.6 |
| Figueiredo et al[43] | COPD (GOLD I–IV) | Examine coping strategies of family cgs of pts with early and advanced COPD. To analyse subjective burden of family cgs of pts with early and advanced COPD and its predictor variables and how those relate to their subjective health | QUANT (correlational) | Portugal | Cross-sectional convenience | 158 (120 F, 38 M) (>4 years) | 58.4 (early COPD) 60.8 (advanced COPD)69.4 |
| Figueiredo et al[44] | COPD (GOLD I–IV) | To analyse subjective burden of family cgs of pts with early and advanced COPD and its predictor variables | QUANT (correlational) | Portugal | Cross-sectional convenience | 167 (125 F, 42 M) (>4 years) | 58.3 (early COPD) 60.5 (advanced COPD)69.3 |
| Figueiredo et al[67] | COPD (moderate to severe) | Obtain knowledge on experience of husbands and sons providing care to a family member | QUAL | Portugal | Interviews purposive | 12 (M) (>4years) | 70.9 (husbands) 43.4 (sons)/72.1 |
| Grigorovich et al[45] | HF (NYHA II–IV) | To examine changes in cg's well-being over time. Identify pt and cg factors associated with positive and negative outcomes | QUANT (repeated measures) | Canada | Longitudinal convenience | 50 (31 F, 19 M) (mean=18 months) | 58/61.6 |

Continued

**Table 2** Continued

| First author/ref. no. | Diagnosis | Aims (as stated by authors) | Methods | Country | Data collection sampling | Participants caregiver (time caring) | Mean age (caregiver/recipient) |
|---|---|---|---|---|---|---|---|
| Halm et al[46 47]* | CAD | To determine cg burden after CABG surgery | QUANT | USA | Cross-sectional convenience | 166 (136 F, 30 M) (≤12 months) | 64.7/66.8 |
| Halm[68 69]* | CAD | To describe the concerns needs, strategies and advice of CABG cgs during the first 3 months postsurgery. To explore cg burden by age and gender | QUAL | USA | Interviews purposive | 32 (16 F, 16 M)(NS) | 60.6 (M<70)/60.1 61.5 (F<70)/62.5 75.9 (M>70)/74.4 73.6 (F>70)/77.6 |
| Hess[86] | HF | To examine the association between cg literacy and medication administration | QUANT (correlational) | USA | Cross-sectional convenience | 5 (F) (NS) | 65/72.8 |
| Hooley[48] | HF (NYHA III or IV) | To explore if greater cg burden is associated with increasing disease burden and depressive symptoms in pts and cg | QUANT | Canada | Cohort study convenience | 50 (40 F, 10 M) (NS) | 61/72 |
| Hwang et al[49] | HF (NYHA I–IV) | To identify factors associated with the impact of caregiving | QUANT | USA | Cross-sectional convenience | 76 dyads (54 F, 22 M) (mean=53.4 months) | 53.4/53.8 |
| Hynes[70] | COPD | To explore the experiences of cgs providing care to a family member with COPD | QUAL (phenomenological) | Ireland | Interviews convenience | 11 (9 F, 2 M) (1–15 years) | NS/NS |
| Imes et al[71] | HF (NYHA III–IV) | To describe cg's experience of living with HF | QUAL (descriptive) | USA | Interviews convenience | 14 (11 F, 3 M) (NS) | 64.8/68 |
| Karmilovich[50] | HF (NYHA III or IV) | To examine cg demands and components of caring. Assess stress and correlation with cg burden. | QUANT (correlational) | USA | Descriptive survey purposive | 41 (30 F, 11 M) (NS) | 56.7/NS |
| Kitko[72] | HF | To gain a deeper understanding of the type of work in spousal caregiving | QUAL | USA | Interviews convenience | 20 (14 F, 6 M) (2 months–9 years) | 67/70 |
| Kneeshaw[87] | CAD | To examine cg mutuality and preparedness for caring post-CABG surgery | QUANT | USA | Longitudinal convenience | 49 (32 F, 17 M) (NS) | 50.1/72.6 |
| Liljeroos et al[73] | HF | To understand perceived caring needs in dyads and understand areas of support for cgs | QUAL | Sweden | Focus groups convenience | 19 dyads (NS) (7 F, 12 M) | 70/72 |
| Lindqvist et al[74] | COPD (mild to severe) | To describe conceptions of daily life for women caring for men with COPD | QUAL (phenomenological) | Sweden | Interviews purposive | 21(F) (NS) | 72/NS |
| Loftus[51] | HF (NYHA II–IV) | Investigate outcomes of caregiving in late-stage HF | QUANT (correlational) | UK | Longitudinal convenience | 53 (41 F, 12 M) (6.66 hours/day) | 66.7/76.3 |
| Lum et al[91] | HF (NYHA II–IV) | Measure of relationship quality and cg benefit, burden and depressive | QUANT (correlational) | USA | Cross-sectional purposive | 19 (7 F, 12 M) (<1 to >8 hours/week) | 59/69 |
| Luttik et al[75] | HF | Experience and needs of cg's well-being factors | QUAL | The Netherlands | Interviews convenience | 13 (10 F, 3 M) (NS) | 66/88.6 |
| Luttik et al[92] | HF (NYHA II–IV) | Investigate QoL in cgs of pts with HF vs QoL in people with healthy partners | QUANT | The Netherlands | Cross-sectional purposive | 303 (NS) | 67/69 |
| Marcuccilli[76] | HF-LVAD in situ | Explore life adjustments of cgs caring for long-term LVAD pts. | QUAL (phenomenological) | USA | Interviews convenience | 5 (5 F) (NS) | 56.6/NS |
| Marcuccilli et al[77] | HF-LVAD as DT | Explore experience of caring for family member with HF | QUAL (phenomenological) | USA | Interviews purposive | 7 (6 F, 1 M) (18–24 hours/day) | 65/NS |

Continued

**Table 2** Continued

| First author/ref. no. | Diagnosis | Aims (as stated by authors) | Methods | Data collection sampling | Country | Participants caregiver (time caring) | Mean age (caregiver/recipient) |
|---|---|---|---|---|---|---|---|
| Miravitlles et al[80] | COPD | Analyse burden of cgs | QUANT | Survey representative (mean=12.7 hours daily, severe COPD) | Spain | 22795 | 56.5/72 |
| Nakken et al[52] | COPD | Investigate differences in male and female cgs and their perception of pts' symptoms | QUANT (correlational) | Cross-sectional convenience | The Netherlands | 188 dyads (103 F, 85 M) (NS) | 65.4/63.3 F 65.1/68.7 M |
| Näsström et al[84] | HF | Cg's participation and perspective of home care services | Mixed methods | Interviews purposive | Sweden | 15 (11 F, 4 M) (NS) | 77/NS |
| Park et al[88] | CAD | Difficulty and demands of cg tasks for older cgs of CABG pts | QUANT | Cross-sectional convenience | USA | 35 (29 F, 6 M) (mean=19 days) | 60/NS |
| Pattenden[78] | HF | Explore how pts and cgs cope with daily life with HF | QUAL | Interviews purposive | UK | 20 (18 F, 2 M) (NS) | 67.8/NS |
| Yeh[53] | HF | Explore burden on family cgs of older pts with HF | QUANT (correlational) | Cross-sectional purposive | USA | 50 (35 F, 1 5M) (<6months to >1year) | 60.3/77.6 |
| Pressler et al[54] | HF (NYHA I-IV) | Examine changes in cg burden and HRQoL. Determine different perceptions between cg's of pts. with HF and to estimate time spent on cg tasks | QUANT (repeated measures) | Longitudinal convenience | USA | 65 (48 F, 17 M) (mean=9.3years) | 59.7/69 |
| Riegner[89] | COPD | To understand QoL and its association with role strain, humour and support in cgs and pts. | QUANT (correlational) | Cross-sectional convenience | USA | 83 dyads (50 F, 33 M) (NS) | 63.2/65.6 |
| Rolley et al[79] | CAD | Describe experience of cgs of pts undergoing percutaneous coronary intervention | QUAL | Focus groups convenience | Australia | 18 (F) (NS) | NS/NS |
| Saunders[55 56] | HF (NYHA I-IV) | To determine indicators of cg HRQoL | QUANT (correlational) | Cross-sectional purposive | USA | 50 (42 F, 8 M) (mean=5.9years) | 58.1/77.6 |
| Saunders[57] | HF | Compare employed and unemployed cgs on depression and well-being | QUANT | Cross-sectional convenience | USA | 41 (37 F, 4 M) (2.9-6years) | 59 (unemployed)/78 52 (employed)/77 |
| Schwarz[58] | HF | Evaluate support on stress for cgs | QUANT | Cross-sectional convenience | USA | 75 (55 F, 20 M) (mean=6years) | 63/NS |
| Scott[59] | HF | HRQoL of cgs and pts receiving community-based inotropic infusions | QUANT | Cross-sectional purposive | USA | 18 (16 F, 2 M) | 63/69.3 |
| Spence et al[80] | COPD (advanced) | Needs and experiences of family cgs | QUAL (descriptive) | Interviews purposive | Northern Ireland | 7 (6 F, 1 M) (1–4 years) | NS/NS |
| Strøm (2015) | HF | Next of kin's experience and responsibilities when caring | QUAL | Interviews convenience | Norway | 19 (17 F, 2 M) (NS) | Median 63/NS |
| Takata et al[60] | COPD | Explore cg burden (long-term O2 therapy) | QUANT | Cross-sectional convenience | Japan | 45 dyads (37 F, 8 M) (NS) | 68/75.2 |
| Vellone et al[61] | HF (NYHA I-IV) | Examine cg self-efficacy and contribution to pt self-care | QUANT (correlation) | Cross-sectional convenience | Italy | 515 dyads (270 F, 245 M) | 56.6/75.6 |
| Wallin et al[82] | CAD | To describe cg's need for support and impact after a cardiac event | QUAL (descriptive) | Interviews purposive | Sweden | 20 (14 F, 6 M) (NS) | 55/NS |

Continued

**Table 2** Continued

| First author/ref. no. | Diagnosis | Aims (as stated by authors) | Methods | Data collection sampling | Country | Participants caregiver (time caring) | Mean age (caregiver/recipient) |
|---|---|---|---|---|---|---|---|
| Woolfe[62] | COPD | Identify needs of cgs and how this impacts cg well-being | QUANT (descriptive) | Cross-sectional convenience | Australia | 63 (39 F, 24 M) (NS) | NS/NS |
| Wingham et al[83] | HF | Identify needs of cgs to inform development of a caregiver resource for use in a home- based self-management intervention | QUAL | Interviews (I) focus groups (FG) (purposive) | UK | 22 (16 F, 6 M) 6 months–8 years | (I) 67/NS (FG) 62/NS |

*Same study.
ADL, activities of daily living; CABG, coronary artery bypass graft; CAD, coronary artery disease; cg, caregiver; COPD, chronic obstructive pulmonary disease; HF, heart failure; pt, patient; HRQoL, health-related quality of life; LVAD DT, left ventricular assist device destination therapy; LVAD, left ventricular assist device; NS, not stated; QoL, quality of life; NYHA, New York Heart Association; GOLD, Global Initiative for Chronic Lung Disease, QUANT, quantitative; QUAL, qualitative.

tasks from their routine.[36 39 59] Contrastingly, Pressler et al reported caregivers' perceptions of how their lives changed as a result of caregiving was neutral and improved from baseline to 4 and 8 months.[54] Caregivers became adaptable in their new role.[72 80 85] There was less personal time for leisure and hobbies either alone or with the care recipient.[67–70 76 77 82 83] Caregivers described daily 'ups and downs' and had to adjust their routines dependent on the presentation of the care recipient.[63 64 66 71 73 75 78 79 81 83]

### Support for caregivers

Fifteen quantitative,[41 45–47 49 51–53 55 56 58–60 62 87 89] 21 qualitative[63–83 85] and 1 mixed methods[84] study examine support. This includes healthcare, family and social support. The weight of perceived external expectations, the necessity of being proactive in obtaining support and maintaining a social role was described across all diagnoses.[45 46 48–50 53–57 61 89] Yeh and Bull identified lack of family support as a significant issue.[53] Caregivers felt abandoned by healthcare teams. After hospital discharge they had to provide care without advice or medical support.[66 72 78] Positive interactions were reported, namely access to healthcare professionals via telephone or home support.[63 64 77 84]

### Knowledge

This was addressed in 17 qualitative,[63 65–75 77 79 80 82 83 85] 5 quantitative[50 60–62 87] and 1 mixed methods[84] study. This describes caregivers' understanding of the diagnosis and need for knowledge throughout the duration of illness.[63 67 70 75 83–85] Caregivers report information from health professionals was often inadequate.[71 73 74] Timing and format of information was significant. Caregivers received information verbally or by leaflets in hospital but describe being left alone to provide care in the long term.[65 68 69 79 82] Caregivers had difficulty understanding how to navigate the care system.[72 80] They had to make decisions without full knowledge of the consequences of their decision making, particularly during acute exacerbations.[65] The quantitative element of mixed methods study by Näsström et al correlated with qualitative studies; receipt of sufficient information was central to managing HF and was associated with better perceived health of caregivers.[84]

### Relationships

Twenty qualitative,[64–83 85] 22 quantitative[35 37 38 42–44 46–52 54–57 61 87 90–92] and 1 mixed methods[84] study examined relationships. In HF studies caring for individuals with more symptoms resulted in poorer perceived experiences.[54 91] Higher relationship quality resulted in less burden and more benefit from the relationship. The relationship prior to diagnosis influenced the current relationship. Perspective of the relationship was either a sense of duty[65 74 80 81 84] or this was a valuable second chance.[66 75 82 83] Caregivers reported difficulty communicating about the condition leading

**Table 3** Study characteristics

| Summary of study characteristics | n=54 studies | |
|---|---|---|
| **Aetiology of patients** | | |
| CAD, n (%) | 6 (11) | |
| HF, n (%) | 34 (63) | |
| COPD, n (%) | 14 (26) | |
| **Caregiver participants*** | | Patients† |
| Median age (range) | 62 (43–77) | 69 (36–93) |
| Median % of female (range) | 63% (5–270) | 38% (1–229) |
| **Relationship between patient and caregiver (n=26 008 caregivers)** | | |
| Spousal/partner, n (%) | 2321 (9) | |
| Son/daughter, n (%) | 610 (2) | |
| Sibling, n (%) | 22 (<1) | |
| Parent, n (%) | 10 (<1) | |
| Friend/relative, n (%) | 228 (<1) | |
| Not stated | 22 961 (88) | |
| **Type of study** | | |
| Qualitative, n (%) | 21 (39) | |
| Quantitative, n (%) | 32 (59) | |
| Mixed | 1 (2) | |
| **Study design** | | |
| Cross-sectional, n (%) | 24 (44) | |
| Longitudinal, n (%) | 4 (7) | |
| Cohort, n (%) | 2 (4) | |
| Quantitative (survey), n (%) | 2 (4) | |
| Qualitative (interview/focus group), n (%) | 16 (31) | |
| Phenomenological, n (%) | 5 (8) | |
| Inductive, n (%) | 1 (2) | |
| **Continents of publication** | | |
| Europe | 22 (41) | |
| North America, n (%) | 29 (54) | |
| Australasia, n (%) | 3 (5) | |
| **Date of publication** | n=57 publications‡ | |
| 1990–1995 | 2 | |
| 1996–2001 | 3 | |
| 2002–2007 | 10 | |
| 2008–2013 | 22 | |
| 2014–2017 | 20 | |

*Caregiver data based on data collected in 50 studies.
†Patient data based on data collected in 35 studies.
‡There were 57 publications, however there were 54 studies. The following studies used the same data but produced two publications: Halm, 2006 and 2007, Saunders, 2008 and 2009, Halm 2016 and 2017.
CAD, coronary artery disease; COPD, chronic obstructive pulmonary disease; HF, heart failure.

to isolation, stress and conflict between caregiver and care recipient.[71 73] The relationship requires negotiation.[69 85] Caregivers prioritised the care recipient over their own needs.[64 72 74 77 82]

### Expert by experience

Twelve qualitative studies[65–70 72 75 76 80 81 83 85] addressed this concept. Caregivers learnt new skills. They became 'experts by experience' discovering through 'doing' and

**Table 4** (a) Quality appraisal—qualitative papers

| First author/ref. no. | Design | Recruitment | Data collection | Data analysis | Findings | Total |
|---|---|---|---|---|---|---|
| Andersen[63] | 1 | 1 | 1 | 0 | 1 | 4 (H) |
| Baker et al[64] | 1 | 0 | 1 | 1 | 1 | 4 (H) |
| Bove et al[65] | 1 | 1 | 1 | 1 | 1 | 5 (H) |
| Burke et al[66] | 1 | 1 | 1 | 1 | 1 | 5 (H) |
| Clark et al[85] | 1 | 1 | 1 | 1 | 0 | 4 (H) |
| Figueiredo et al[67] | 0 | 1 | 1 | 1 | 1 | 4 (H) |
| Halm[68]* | 1 | 1 | 1 | 1 | 1 | 5 (H) |
| Halm[69]* | 1 | 1 | 1 | 1 | 1 | 5 (H) |
| Hynes[70] | 1 | 1 | 1 | 0 | 1 | 4 (H) |
| Imes et al[71] | 1 | 1 | 1 | 1 | 1 | 5 (H) |
| Kitko[72] | 1 | 1 | 1 | 1 | 1 | 5 (H) |
| Liljeroos et al[73] | 1 | 1 | 1 | 1 | 1 | 5 (H) |
| Lindqvist et al[74] | 1 | 1 | 1 | 1 | 1 | 5 (H) |
| Luttik et al[75] | 1 | 1 | 1 | 1 | 1 | 5 (H) |
| Marcuccilli[76] | 1 | 0 | 1 | 1 | 1 | 4 (H) |
| Marcuccilli[77] | 1 | 1 | 1 | 1 | 1 | 5 (H) |
| Näsström et al[84]† | 1 | 1 | 1 | 1 | 1 | 5 (H) |
| Pattenden[78] | 1 | 1 | 1 | 1 | 0 | 4 (H) |
| Rolley et al[79] | 1 | 1 | 1 | 1 | 0 | 4 (H) |
| Spence et al[80] | 1 | 0 | 1 | 1 | 1 | 4 (H) |
| Strøm (2015) | 1 | 1 | 1 | 1 | 1 | 5 (H) |
| Wallin et al[82] | 1 | 0 | 1 | 1 | 1 | 4 (H) |
| Wingham et al[83] | 1 | 1 | 1 | 1 | 1 | 5 (H) |

*Same study.
†Mixed methods study—qualitative component.
H, high quality, 4/ 5 out of 5 quality criteria achieved.

observing health professionals.[66 68 83] They developed 'proto-professional skills'; in medication administration[65 80 85] judging care recipients' level of functioning[79] and decision making in times of exacerbations.[70] Caregivers observed the nuances of change in the care recipient often not perceived by healthcare teams or other family members such as skin colour or irritability.[72 75 81]

### Vigilance
Vigilance was recurring in caregivers' narrative across all diagnoses and was present in 19 qualitative studies.[64–81 83 85] Caregivers were always on the alert observing the care recipient.[66 67 70 72–74 77–79 81] They lay awake at night listening for their partners' breath.[69 71 75 85] This impacted on caregivers' health creating constant fatigue, worry and stress.[65 79] Caregivers recognised that the need for vigilance came from themselves and their insecurities.[64 76 83]

### Time
Time explores how caregivers adjusted to living with the illness and was present in 15 qualitative studies.[65 67–77 80 82 83 85] Caregivers adapted to a new life, referring to 'then', how life was and 'now' their current

life.[69 70 75 76 83] The duration of caregiving and severity of illness influenced caregiver's ability to adjust.[66 73 76] Caregivers lived day by day[83] and viewed the future, with hope or uncertainty about what lay ahead.[65 70 72 79 82]

### Shared care
Shared care was present in 16 qualitative studies.[63–66 68–76 80 81 83 85] This demonstrates caregiver and care recipient working together managing the illness, jointly administering medication[68 81] and attending appointments.[73] The presence of illness was a process they adjusted to together.[76 80] Caregivers referred to themselves and the care recipient as 'we', when discussing dealing with the illness.[63 71 75] The mutual perspective between caregiver and care recipient served to isolate them from the world, the illness was *'taking a life of its own…it's like this third person'* (Hynes, 2012, p. 1071).

There were differences in caregiver experience for each of the diagnoses and these are discussed below.

### Heart failure
HF caregivers experienced an 'ebb and flow' in caring, an underlying worry, fear and anxiety, which at times of

**Table 4** (b) Quality appraisal—quantitative papers

| First author/ref. no. | Study design | Participant sampling | Participant attrition | Outcome measures | Data analysis | Overall score |
|---|---|---|---|---|---|---|
| Ågren[37] | CS | Purp (+1) | 0% (+1) | Non-V | MV (+1) | 3 (M) |
| Al-Rawashdeh[38] | CS | Purp (+1) | NS | V (+1) | MV (+1) | 3 (M) |
| Badr et al[39] | CS | Con | 15.5% (+1) | Non-V | MV (+1) | 2 (L) |
| Bakas et al[40] | CS | Con | NS | V (+1) | MV (+1) | 2 (L) |
| Chung et al[36] | CS | Con | NS | V (+1) | UV | 1 (L) |
| Cossette[41] | CS | Con | NS | V (+1) | MV (+1) | 2 (L) |
| Evangelista et al[42] | CS | Con | 20% (+1) | V (+1) | MV (+1) | 3 (M) |
| Figueiredo et al[43] | CS | Con | 17% (+1) | V (+1) | MV (+1) | 3 (M) |
| Figueiredo et al[44] | CS | Con | 11% (+1) | Non-V | MV (+1) | 2 (L) |
| Grigorovich et al[45] | LS (+1) | Con | NS | V (+1) | MV (+1) | 3 (M) |
| Halm et al[46]* | CS | Con | 64% | V (+1) | MV (+1) | 2 (L) |
| Halm[47]* | CS | Con | 64% | V (+1) | MV (+1) | 2 (L) |
| Hess[86] | CS | Con | NS | V (+1) | MV (+1) | 2 (L) |
| Hooley[48] | CS | Con | 0% (+1) | V (+1) | UV | 2 (L) |
| Hwang et al[49] | CS | Con | 35% | V (+1) | MV (+1) | 2 (L) |
| Karmilovich[50] | CS | Purp (+1) | 24% | V (+1) | MV (+1) | 3 (M) |
| Kneeshaw[87] | LS (+1) | Con | 32.70% | V (+1) | MV (+1) | 3 (M) |
| Loftus[51] | LS (+1) | Con | 36% | V (+1) | MV (+1) | 3 (M) |
| Lum et al[91] | CS | Purp (+1) | 5% (+1) | V (+1) | MV (+1) | 4 (H) |
| Luttik et al[92] | CS | Purp (+1) | 31% | Non-V | MV (+1) | 3 (M) |
| Miravitlles et al[90] | CS | Rand (+1) | 0% [+1] | Non-V | MV (+1) | 3 (M) |
| Nakken et al[52] | CS | Con | 58% | Non-V | MV (+1) | 1 (L) |
| Näsström et al[84]* | LS (+1) | Purp (+1) | 7% (+1) | V (+1) | MV (+1) | 5 (H) |
| Park et al[88] | CS | Con | NS | V (+1) | UV | 1 (L) |
| Yeh[53] | CS | Purp (+1) | 39% | V (+1) | MV (+1) | 4 (H) |
| Pressler et al[54] | LS (+1) | Con | 16% (+1) | V (+1) | MV (+1) | 4 (H) |
| Riegner[89] | CS | Con | 71.80% | V (+1) | MV (+1) | 2 (L) |
| Saunders[55]† | CS | Purp (+1) | 36% | V (+1) | MV (+1) | 3 (M) |
| Saunders[56]† | CS | Purp (+1) | 36% | V (+1) | MV (+1) | 3 (M) |
| Saunders[57] | CS | Con | NS | V (+1) | UV | 1 (L) |
| Schwarz[58] | CS | Con | NS | V (+1) | MV (+1) | 2 (L) |
| Scott[59] | CS | Purp (+1) | 10% (+1) | Non-V | MV (+1) | 3 (M) |
| Takata et al[60] | CS | Con | NS | V (+1) | UV | 1 (L) |
| Vellone et al[61] | CS | Con | NS | V (+1) | MV (+1) | 2 (L) |
| Woolfe[62] | CS | Con | 37% | V (+1) | UV | 1 (L) |

Studt design: CS, LS.
Participant sampling: Purp, Rand, Cons, Con, NS.
Attrition: 20% or less=+1; NS.
Outcome measures: V, non-V, NS.
Data analysis: MV, UV.
*Same study.
†Mixed methods study—quantitative component.
CS, cross-sectional design; Cons, consecutive; Con, convenience; H, high quality, 4/ 5 out of 5 quality criteria achieved; L, low quality, 1 or 2 out of 5 quality criteria achieved; LS, longitudinal design; M, medium quality, 3 out of 5 quality criteria achieved; MV, multivariate; non-V, some or all non-validated; NS, not stated/unclear; Purp, purposive; Rand: random; V, all validated outcomes; UV, univariate.

change or illness heightened.[49 51 59 71 77 81 83 85 92] Lifestyle changes were long lasting and sustained.[39 59 64 71 77 81 83 85 92] Obtaining knowledge was necessary throughout all stages of the illness.[50 63 66 73 83 85 92] Sourcing information and communication with health professionals was often difficult.[63 66 71 85 92] In spousal relationships, they predominantly

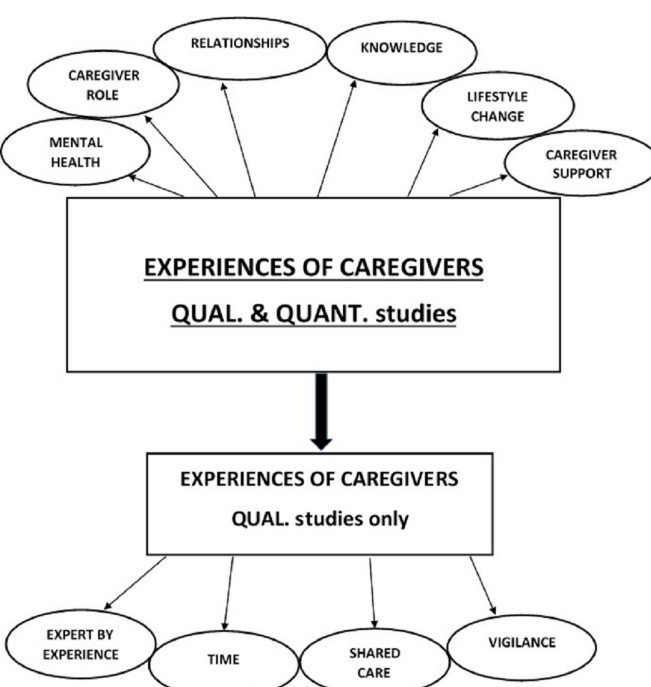

**Figure 2** Conceptual experiences of caregivers.

viewed the care recipient as another child or as a 'duty'.[50 51 64 66 71 73 77 84]

### Chronic obstructive pulmonary disease

COPD caregivers experienced a prolonged impact on their mental health similar to HF caregivers.[41 44 52 60 65 70 80] Severity of illness was influential on their experience of burden.[38 43 44 60] Role change was long lasting and profound for many.[65 67 70 80] They expressed concerns with perceived lack of knowledge.[62 65 70 74 80] During exacerbations, COPD caregivers experienced anxiety and fear of their loved one dying.[65 67 70 74] COPD caregivers highlighted the loss of social roles while trying to maintain the dignity of their loved ones.[65 70 74 80 89 90] The coughing and spitting associated with COPD often left the care recipient embarrassed.[65 67 80] The caregiver tried to avoid situations where this would happen. The dynamics of spousal relationships changed, caregivers described losing the intimate love they had for their partner.[65 70 74 80]

### Coronary artery disease

Caregivers of patients with CAD experienced intense role change on discharge from hospital and in the acute phase of illness.[47 68 79 82 87 88] They initially engaged with a high volume of tasks which reduced over time.[46 47 68 79 82 87 88] CAD caregivers experienced post-traumatic symptoms if they witnessed the recipient experience an acute event.[79 82] Caregivers described being unable to talk about this and reliving the event in their heads. Anxiety did ease over time for many.[79 82] Caregivers felt unprepared at hospital discharge and highlighted not realising how much their routine would be disrupted.[68 79 82] Caregivers reported viewing the experience as a second chance and

had a renewed sense of love and appreciation for the relationship.[79 82]

### DISCUSSION

This mixed methods systematic review demonstrates the similarities and differences in caregiver experiences across three common cardiorespiratory conditions. It highlighted the differences in experiences obtained from qualitative and quantitative research. Commonly occurring experiences included the exacerbation of caregiver physical and mental health due to the role. This correlates with systematic review of HF caregivers by Kang *et al* identifying that caregiving resulted in a multitude of changes in caregiver's lives regardless of age, gender and ethnicity.[93] Addressing both patient and caregiver needs in order to maintain well-being for both is important[19] and recognises the value of 'shared-care' between patient and caregiver. The prevalence of mental health needs in this review demonstrates the need for psychosocial support for caregivers. This concurs with the studies by AasbØ *et al*, identifying caregivers need to be in 'emotional control'[94] and Wingham *et al*, describing the 'enduring anguish' experienced by caregivers.[95] Lawton *et al* attribute caregiver well-being to the commitment of the caregiver to the role and dealing with competing demands, which can increase burden and negatively impact affect. Spousal caregivers may be more ready to accept the role of caregiving than adult children who may view it as an imposition on an already established lifestyle.[96] In this review, societal expectations had an impact on how caregivers adjust to their role. Additionally, the quality of the relationship prior to becoming a caregiver had an influence on the caregiver subjective experience of burden.

Caregivers had predominantly negative experiences of support and described uncertainty of how to obtain this. Caregivers need greater support and knowledge transfer to conduct their role.[97] They should be included in clinical appointments[98] to ensure they are not isolated in providing care and to allow for knowledge exchange. Giacomini *et al* in their review of caregivers living and dying with COPD reported increasing isolation in addition to their own health issues.[18] They described pressure balancing their variety of roles; similar experiences to the caregivers in this synthesis across all diagnoses. Caregivers emphasised their need to be vigilant. This falls into five categories as defined by Mahoney's study of Alzheimer's caregivers; 'watchful supervision', 'protective intervening', 'anticipating', 'on duty' and 'being there'. Caregivers in this synthesis described overt vigilance, putting one's head on the chest of the recipient to check breathing or covert vigilance; observing them throughout the day.[99] Healthcare professionals must be aware of these levels of vigilance and the constant presence of them to support caregivers in their role.

Caregivers are valuable providers of care. Caregiver's needs should be assessed systematically and in a formalised manner in healthcare settings.[16] When

**Table 5** Illustrative quotes of caregiver experience—by concept

| | |
|---|---|
| Mental health | "The mental strain is difficult. I feel so trapped". <br> "You fall into a huge hole, then the world gets so tiny, it all gets sonarrow that it is almost unbearable". <br> "I feel like sleeping beauty. The hawthorn hedge has closed around me, and I cannot do anything about it.". |
| Role | "I can sum my role up in three words, I am a cheerleader, drill sergeant, and negotiator". |
| Lifestyle change | "Our life has come down. The two of us used to go out dancing. We loved dancing and then it all stopped". |
| Knowledge | "I wish I had had more education on the 'what ifs'. When I was leaving the hospital nobody really said, 'OK now this is what's going to happen and this is what you're going to have to do'. If there would've been any kind of complications I would've been totally in the dark. I didn't know all the things I needed to know". |
| Relationships | "I just love him and I find that every day when I see him, what else could I do to try and make him a wee bit … better? It's very satisfying to know that he appreciates what I do and it's nice to know that you are helping someone". <br> "It's like having another child sometimes because you are sort of responsible and I feel he is my responsibility. I feel that he is not anybody else's responsibility…" |
| Support | "And then I really felt alone in it all. Because everybody would call and come over and ask, how is John? Hardly anyone asked 'how are you doing'"? <br> "Doctors (do) not realize that 1 day your life is jut normal and then this comes and smashes everything to bits, you know and there are so many questions". <br> "I would be lost without, our heart failure nurse, and, all the other input we've had from all the other professionals, like the podiatrist and GP… You can do it, but in partnership with everybody else". |
| Vigilance | "Every morning I put my ear to his chest and listen to his heart, that is how we first discovered he was in atrial fibrillation so now I do it every morning before I leave. I monitor him very closely and there are days in which I do not feel comfortable leaving for work so those days I work at home. I call everyday from work and we have our routine, if I am not aware of anything he had planned for the day, I then immediately call my neighbour to check on him". |
| Shared care | "There were days I thought to myself, where are we going from here? But we mastered it together and tried to do things at his pace". |
| Time | "At first it was overwhelming. I didn't think I could do it. When they first told me I was like, 'I can't do that', you know. And then they explained to me, like, yes you can. It's like getting a new baby. You know, you learn how to take care of them step by step and then it's just part of the routine. And that's really the way it was". |
| Expert by experience | "It's so frustrating when she goes into hospital and the nurses and the doctors say it's her condition, you know. I'm like I'm with her twenty-four hours a day, I know how breathless she is without infection and I know how breathless she is with an infection there's a major difference". <br> "I see him every day, they are just little subtle changes, they are not showing up in the numbers the doctors are concerned with but I see it". |

developing collaborative models of care the inclusion of caregivers is imperative.[100]

## Strengths and limitations

This review demonstrates the complexity of what it means to be a caregiver and should inform clinical care development of interventions. A mixed methods review can be contentious[101] due to the synthesis of differing paradigms. In this review, it required transformation of quantitative data into qualitative data.[15 102] We aimed to present a convergence of caregiver experiences by conducting a mixed methods synthesis. However, it demonstrated four differing concepts between the two paradigms. This highlights the challenge of synthesising multiple methods. It is worth exploring how the four additional qualitative concepts could be captured quantitatively in order to inform healthcare intervention. This mixed methods synthesis is, to our knowledge, the first to

combine caregiver experiences in HF, COPD or CAD. It examines the differences and similarities in experiences, establishing a comprehensive assessment of the knowledge base of caregiver experiences in common cardiorespiratory conditions.

There are limitations to this study; both in our review methods and the nature of included studies. First, we acknowledge that the inclusion of lower quality quantitative studies may lead to risk of bias: the majority of quantitative studies used convenience sampling, had a high attrition and low response rate. Non-validated outcome measures were used in some quantitative studies with the majority of studies conducting univariate rather than multivariate analysis. However, given the limited number of high-quality quantitative studies (four studies), we believe this broader inclusion increased the scope of our review in order to achieve a holistic understanding of

caregiver experiences. Furthermore, we would note that the conclusions of this review were broadly the same with consideration of only the high-quality quantitative studies. Second, studies were restricted to English language only, from high-income countries and excluding caregivers of nursing home residents. This may limit the applicability of findings to other settings. Third, converting quantitative data into a qualitative data set risks the quantitative data set being oversimplified. This was managed with regular research team meetings to review each stage of this process. Fourth, qualitative synthesis is an interpretation of other researcher's interpretations. To minimise individual interpretative bias, a second researcher was used to seek confirmation of the results. Finally, included qualitative and quantitative studies were primarily cross-sectional in design, therefore considering caregiver experience only at a single point of time.

## Implications for practice and future research

There are a number implications following this review. It has demonstrated there are similarities and differences in the caregiver experience in HF, COPD or CAD. The impact on caregiver's lives of those with HF and COPD appears longer lasting and more turbulent than caring for patients with CAD. CAD caregiver's experience of hospital during exacerbations increased distress at discharge. This review reflects the complexity of the caregiver's role. The mixed methods approach indicted differences in what is being investigated. This is important in demonstrating an understanding of the caregiver experience when dealing with complex conditions. Future research should focus on involving caregivers in the design and delivery of interventions for patients with cardiopulmonary disease. Best practice interventions for CAD caregivers in the discharge process from hospital to home must be formalised. There appears to be a focus on the mental health of caregivers of those with HF; however, further research is needed to explore this in COPD and CAD caregivers. Exploration of this via support groups for caregivers of cardiorespiratory conditions is merited. Clinically, the healthcare team need to identify who the caregiver is and be aware of their needs with the use of a carer's needs assessment. There must be a greater understanding of caregiver support needs, what they avail of and are they aware of what is available to them in the community. This can be achieved in conversation between the healthcare team and caregivers and warrants further research as to how and whether caregivers avail of external supports.

Consideration needs to be given as to whether quantitative research tools to explore caregiver expertise, view of the future, experience of shared care and vigilance can be developed to capture these qualitative concepts to inform the development of self-management interventions for patients and caregivers. Repeated measures examining perceived control and caregiver needs may contribute to a greater understanding of caregiver experiences, which arose in qualitative studies. Additionally, longitudinal studies with repeated assessment need to be conducted to assess the stability of caregiver experiences and whether they are liable to much change over time. In this review, only 4 out of 32 quantitative studies examined caregiver's experiences longitudinally. Understanding whether there are caregiver changes over time will facilitate greater understanding of caregiver needs for health professionals when working with this population. The emergence of additional concepts from qualitative studies emphasises the role of mixed methods research when examining lived experiences. The additional concepts also demonstrated the nuanced expertise of the caregiving experience. It is important for researchers to consider how to reflect this in quantitative investigation so as to inform funders in order to develop and trial interventions in HF, COPD and CAD. The quality of quantitative studies in COPD and CAD were medium or low. There is a need for more empirically robust studies examining the experiences of these caregivers. Additionally, greater understanding of caregiver's experiences with this population will facilitate the development of robust evidence-based guidelines for health services when working with HF, COPD and CAD.

## CONCLUSIONS

This mixed methods systematic review provides a holistic synthesis of caregiver experiences of people with HF, COPD or CAD. It demonstrates there are a number of implications when an individual becomes a caregiver for those with chronic cardiopulmonary disease. Caregivers reframe their identity and change their life course. Caregivers learn a multitude of skills and develop expertise in their new role. Their expertise is invaluable and should be acknowledged in healthcare interventions for these conditions. The quality of evidence was limited by assessment of caregiver experience at single time point. There is need for future studies that employ longitudinal designs examining the change in caregiver experience over time. Caregiving can be positive if caregivers have access to support, are well informed and part of the healthcare team. Understanding the experiences of caregivers for people with these conditions allows healthcare professionals and policy makers to reflect on our approach. Health services must consider caregivers in the design and delivery of interventions.

**Acknowledgements** The authors would like to thank the PenCLARHC Evidence Synthesis Team at University of Exeter Medical School. Katy Oaks and Catriona Organ, Librarians at Royal Cornwall Hospital Library, Knowledge Spa, Truro. Dr Ruth Garside, Senior Lecturer in Evidence Synthesis, European Centre for Environment and Human Health for their assistance throughout the process of this review.

**Contributors** MN, JW, RST: conceived and designed the review and quality appraised the included studies. MN wrote the initial draft of the manuscript. JW and RST provided feedback and edits to manuscript drafts.

**Funding** This study was supported by a University of Exeter Postgraduate Studentship Grant.

**Competing interests** None declared.

**Patient consent** Not required.

**Provenance and peer review** Not commissioned; externally peer reviewed.

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
