## [Reviewer comments · BMJ Open]

ARTICLE DETAILS

TITLE (PROVISIONAL)	"Who Cares?" The Experiences of Caregivers of Adults Living with Heart Failure, Chronic Obstructive Pulmonary Disease and Coronary Artery Disease – A Mixed Methods Systematic Review
AUTHORS	Noonan, Miriam; Wingham, Jennifer; Taylor, Rod

VERSION 1 – REVIEW

REVIEWER	Dr Elise Mansfield University of Newcastle, Australia
REVIEW RETURNED	18-Dec-2017

GENERAL COMMENTS	Overall comments While this is an important topic, I had several significant questions and concerns about the methodology which limited my ability to review and interpret the results. I had some suggestions which I believe may increase the value of the review approach. Is the research question or study objective clearly defined? It is unclear why only HF and CAD were selected for inclusion when the search terms included much broader terms, including cardiomyopathy and stroke. The rationale should include some detail about the symptom burden of each of the examined diseases to justify why it is important to examine carers' experiences. Is the study design appropriate to answer the research question? One of the reasons provided for selecting the mixed methods approach is "to gain a qualitative assessment of the quantitative predictors of experiences of caregivers." It is unclear why only the predictors of experiences would be important to examine? Are the methods described sufficiently to allow the study to be repeated? Given that the experience-related terms are very broad, a more detailed description of these needs to be given in text to give an idea of the scope. Also, "routine" does not appear to be in the list of search terms used. There did not appear to be any cross-checking of the screening of papers against the eligibility criteria. This may have resulted in biases in the selection of studies and should be noted as a limitation to the review. 'Caregiver' needs to be more fully defined, e.g. does it encompass providing practical support only, or emotional support etc? The components of data extraction which is described in text should be more in line with the components included in Table 2. The criteria for assessing methodological quality of quantitative papers were unclear. What standard did the paper have to meet on
---

	each criteria to be scored 1? (e.g. how was appropriate sample recruitment defined?). Were papers of only high methodology quality considered in the review? It also appears that important methodological information is not considered (e.g. response rate, sample size). There are methodological criteria available for assessing methodological criteria of quantitative studies, e.g. Effective Public Health Practice Project (EPHPP) Quality Assessment Tool for Quantitative Studies (Thomas, Ciliska, Dobbins, & Micucci, 2004), or the rating scale developed by Fowkes & Fulton (1991). My preference would be for the authors to utilise a more accepted tool to examine study quality. The narrative profile approach presented in Table 1 needs to be explained in much more detail in the text. It was not clear what each of the profiles and explanations meant, and therefore the meaning and relevance of the information presented in the final column was lost. Is the information presented in the final column an example from one study only? It is also unclear how the concepts listed in supplementary file 2a and 2b were derived from this process. My concern with this process is that it appears to ignore the fact that qualitative and quantitative data are very different. This is supported by the fact that each type of data generated different themes. By forcing the quantitative data into a qualitative framework this misses important information that quantitative data is able to give us, such as the prevalence of specific experiences, and fails to acknowledge the (usually) greater sample representativeness of quantitative studies. The review approach may be strengthened and provide an increased insight into caregiver experience by treating quantitative and qualitative data separately. This would provide additional information about prevalence giving some indication of the relative frequency of different experiences. The eligibility criteria that patients had to be community dwelling (page 31, para 2) is not described in the Method. Do the results address the research question or objective? Given the lack of clarity in reporting of methods it is not possible to determine this. Are they presented clearly? The PRISMA flow diagram is difficult to follow. The direction of the arrow for the number of excluded papers at each step should be reversed. It says there were 7149 excluded due to duplication, but then in the next step 7288 (presumably including these duplicates) which were excluded as they were irrelevant- the duplications should only be counted once. It appears conference presentations were excluded but this was not described in the method. There also seems to be some discrepancies in numbers at all steps of the process- e.g. the number of articles retrieved + articles retrieved from other sources should be equal to the number of studies included in the title and abstract review + number of duplicates excluded, but there is a difference of 2. Please check the additions again. Table 2- The descriptions of the primary focus are very brief and as a result some did not make sense- e.g. Hooley (2005)- "Cg burden, disease severity and depression in pts and cg's"; Hess (2009)- "demographic variable of health literacy"?. It would be better if a more complete description of the aims were presented. It would also be informative if a brief summary of the results of each study were presented in this table. There is also inconsistent use of Cg/cg/cg. There are also abbreviations which have not been defined – e.g. pt., mgt., LVAD, NOK, CABG. Please either spell out if only single use
--	--

	or define abbreviation at end of table. Table 3- why are there 57 publications instead of 54 in the breakdown by date of publication? Also the mean age is described in the text, while this is presented as median in the table. On page 18 it states that if low or medium quality studies which produced a different insight or value to high quality studies then these were included. This is very subjective and there is almost no point completing a quality appraisal of papers if low quality papers are to be included based on their findings. It is difficult to comment on the findings presented on page 22 as it is unclear how these themes were arrived at. In the results sections starting on page 25 it should be identified which are qualitative and which quantitative studies. Also it is not clear which of these studies looked at which type of illness. It would be most informative if the results were presented by theme, and then within each of these by disease. Are the discussion and conclusions justified by the results Page 29- "All caregivers descried not knowing who to contact"- appears inaccurate to say this? The first para of the discussion appears to be a repetition of the results. Greater interpretation is needed and it may be more effective to integrate some of the references included on page 30. All of the data provided from para 2 of page 29 is new and should be presented as part of the results rather than discussion. Why would it be suggested that guidelines be developed for engaging with caregivers of CAD patients only? Unclear what is meant by "Qualitative research clearly demonstrates the caregiving experience is constantly evolving and this needs to be reflected in quantitative research" Are the study limitations discussed adequately? The final point of the study limitations section is more a limitation of the included studies rather than a limitation of the review. A separate section dedicated to the methodological quality and limitations of included studies is warranted. Minor points In the Abstract, the eligibility criteria for selecting studies- should this be "providing care for patients with HF, COPD or CAD"? First sentence of introduction is awkward to read- please consider revising. "The Institute of Public Care examining a Caregivers' Life (2017)"quote in introduction- unclear what this document is. It also needs to be included in the list of references. It would be helpful to have more explanation of the "transitions and the dynamic process of caring" which you mention in the introduction. The web addresses provided in the Methods section should instead be provided in the reference list. In Study quality assessment- "This contributes to a synthesis of evidence which will facilitate obtaining a summary of evidence."- these two things mean that same thing? Typo in Table 1- "Holistic"
--	--

	Some inconsistent use of caregiver/carer throughout Overall more editorial work and proofing is needed.
REVIEWER	Florence Potier CHU UCL Namur, Institute of Health and Society
REVIEW RETURNED	08-Jan-2018
GENERAL COMMENTS	The explores a very interesting topic. The study methods are appropriate. However, I have some suggestions to improve the clarity of the presentation. Results:  1. Table 2: could you add information's about the mean age of the care-receiver and how long the caregiving has occurred? These patient data help clinicians in interpreting the results or implement them in clinical practice. 2. Table 3 : could you add the different designs (to state this important limitation) 3. Table 3: The classical presentation is "CAD, n (%) 6 (11.1)" 4. Figure 2 can be improved (groups with intersections for example. The arrow between "qual and quant papers" and "qual papers only" is not clear.) Discussion:  1. The summary of the findings by diagnosis should be placed in the results section (or in any case before the first sentence: "this mixed methods...condition".) 2. Please, add some "Implications for practice" in the section "implication for practice an future research" (new ideas should not be only in conclusions) I hope the authors will find these comments useful

VERSION 1 – AUTHOR RESPONSE

Reviewer: 1

Reviewer Name: Dr Elise Mansfield

Institution and Country: University of Newcastle, Australia Competing Interests: None declared

Dear Dr. Mansfield.

Many thanks for reviewing this paper and for your helpful comments. I have responded to each of them below.

Overall comments

While this is an important topic, I had several significant questions and concerns about the methodology which limited my ability to review and interpret the results. I had some suggestions which I believe may increase the value of the review approach.

Is the research question or study objective clearly defined?

It is unclear why only HF and CAD were selected for inclusion when the search terms included much broader terms, including cardiomyopathy and stroke. –

Cardiomyopathy has been identified in studies in which heart failure population were included, it was particularly relevant for studies which examined younger patients. If cardiomyopathy had not been included then these studies would have been missed.

Given the substantive body of literature on stroke carers needs and our perception of the differences in needs of stroke patient's vs cardiac diagnoses, we decided it appropriate to exclude

The rationale should include some detail about the symptom burden of each of the examined diseases to justify why it is important to examine carers' experiences.

Further information regarding caregiver burden in these conditions has been included, please see p.4, paragraph 3.

Is the study design appropriate to answer the research question?

One of the reasons provided for selecting the mixed methods approach is “to gain a qualitative assessment of the quantitative predictors of experiences of caregivers.” It is unclear why only the predictors of experiences would be important to examine? – By far and away the main approach to assessing carer's needs in the included quantitative studies was to use predictive/association analyses. So hence we used this data to inform this reviews.

Are the methods described sufficiently to allow the study to be repeated?

Given that the experience-related terms are very broad, a more detailed description of these needs to be given in text to give an idea of the scope. Also, “routine” does not appear to be in the list of search terms used – Amended this as per your comments, please see p. 5, fourth paragraph, under sub-heading “Search Strategy”

There did not appear to be any cross-checking of the screening of papers against the eligibility criteria. This may have resulted in biases in the selection of studies and should be noted as a limitation to the review. – This was conducted. It is now included in the text. Please see p.5, final paragraph under sub-heading “Study selection”.

‘Caregiver’ needs to be more fully defined, e.g. does it encompass providing practical support only, or emotional support etc? – I have elaborated on this point as per your suggestion. Please see p.4, paragraph 1.

The components of data extraction which is described in text should be more in line with the components included in Table 2. – Table 2 has now been amended to reflect this, it now includes aims, key conclusions, time spent caregiving and the mean age of care recipient. Please see p. 12 - 24

The criteria for assessing methodological quality of quantitative papers were unclear. What standard did the paper have to meet on each criteria to be scored 1? (e.g. how was appropriate sample recruitment defined?). Were papers of only high methodology quality considered in the review? It also appears that important methodological information is not considered (e.g. response rate, sample size). There are methodological criteria available for assessing methodological criteria of quantitative studies, e.g. Effective Public Health Practice Project (EPHPP) Quality Assessment Tool for Quantitative Studies (Thomas, Ciliska, Dobbins, & Micucci, 2004), or the rating scale developed by Fowkes & Fulton (1991). My preference would be for the authors to utilise a more accepted tool to examine study quality. – These quality criteria tools were considered and discussed in detail within

the research team. However, due to the design of the studies which were included for synthesis in this review it was identified that a bespoke tool would need to be developed in order to capture the appropriate criteria and key issues for this review. The above mentioned tools guided the development of the bespoke tool that was utilised for this review.

The narrative profile approach presented in Table 1 needs to be explained in much more detail in the text. It was not clear what each of the profiles and explanations meant, and therefore the meaning and relevance of the information presented in the final column was lost. Is the information presented in the final column an example from one study only? It is also unclear how the concepts listed in supplementary file 2a and 2b were derived from this process. – Amended this as per your suggestion. The narrative profile approach is now discussed in more detail on p. 7. Additionally, amended Table 1 to clarify this approach and how it was applied in this review. Table 1 is on p.8 & 9

My concern with this process is that it appears to ignore the fact that qualitative and quantitative data are very different. This is supported by the fact that each type of data generated different themes. By forcing the quantitative data into a qualitative framework this misses important information that quantitative data is able to give us, such as the prevalence of specific experiences, and fails to acknowledge the (usually) greater sample representativeness of quantitative studies. The review approach may be strengthened and provide an increased insight into caregiver experience by treating quantitative and qualitative data separately. This would provide additional information about prevalence giving some indication of the relative frequency of different experiences. – In this context, gathering data using both quantitative and qualitative methods produces a range of results, which are subsequently triangulated to provide an overarching finding. Tashakkori and Teddlie (1998) discuss the pragmatist approach when mixing methods. It considers “what works”, the research question is considered more important than the method or the approach that underlies the method. Johnson et al. (2007) described mixed methods research as a “practical synthesis”, they report it is “a powerful third paradigm choice that will often provide the most informative, complete, balanced and useful research results” p.129. Guided by these principles and a systematic review conducted by Hong et al. (2017) describing how to conduct mixed methods synthesis, the decision was taken to combine the results of this review to produce an overarching narrative to describe caregivers experiences.

References:

Tashakkori, A, Teddlie, C. *Mixed Methodology: Combining Qualitative and Quantitative Approaches*. Applied Social Research Methods Series. 1998; 46. SAGE Publications, Thousand Oaks, CA.

Johnson, R.B., Onwuegbuzie, A.J., Turner, L.A. Toward a definition of mixed methods research. *Journal of Mixed Methods Research* 2007;1: 112–133

Hong et al. *Systematic Reviews* (2017) 6:61. Available from: DOI 10.1186/s13643-017-0454-2

The eligibility criteria that patients had to be community dwelling (page 31, para 2) is not described in the Method. – This has been amended and is now included in study selection, p.5, last paragraph

Do the results address the research question or objective?

Given the lack of clarity in reporting of methods it is not possible to determine this.

Are they presented clearly?

The PRISMA flow diagram is difficult to follow. The direction of the arrow for the number of excluded papers at each step should be reversed. It says there were 7149 excluded due to duplication, but then

in the next step 7288 (presumably including these duplicates) which were excluded as they were irrelevant- the duplications should only be counted once. It appears conference presentations were excluded but this was not described in the method. – The PRISMA diagram has now been updated, please see p.11

There also seems to be some discrepancies in numbers at all steps of the process- e.g. the number of articles retrieved + articles retrieved from other sources should be equal to the number of studies included in the title and abstract review + number of duplicates excluded, but there is a difference of 2. Please check the additions again. – The PRISMA diagram has now been amended, please see p.11

Table 2- The descriptions of the primary focus are very brief and as a result some did not make sense- e.g. Hooley (2005)- “Cg burden, disease severity and depression in pts and cg’s”; Hess (2009)- “demographic variable of health literacy”?. It would be better if a more complete description of the aims were presented. It would also be informative if a brief summary of the results of each study were presented in this table. There is also inconsistent use of Cg/cg/cg. There are also abbreviations which have not been defined – e.g. pt., mgt., LVAD, NOK, CABG. Please either spell out if only single use or define abbreviation at end of table. –Amended Table 2 to make it clearer to follow and included aims and key conclusions of studies. Also updated the legend at the bottom of the table to include any abbreviations that may be present in the table. Please see p.12-24

Table 3- why are there 57 publications instead of 54 in the breakdown by date of publication? Also the mean age is described in the text, while this is presented as median in the table. – Some of the included studies used the same data but were published in different years. This explanation is now included in the legend below Table 3. Please see p. 26/27. RE: median age this is now corrected in the text – please see p. 25 under sub-heading study characteristics

On page 18 it states that if low or medium quality studies which produced a different insight or value to high quality studies then these were included. This is very subjective and there is almost no point completing a quality appraisal of papers if low quality papers are to be included based on their findings.- No low quality qualitative studies were included in this review, therefore, this sentence has been removed.

It is difficult to comment on the findings presented on page 22 as it is unclear how these themes were arrived at.

In the results sections starting on page 25 it should be identified which are qualitative and which quantitative studies. Also it is not clear which of these studies looked at which type of illness. It would be most informative if the results were presented by theme, and then within each of these by disease. – Each paragraph in the results section has been amended to reflect the number of quant. and qual. studies in which each concept emerged. Additionally, the end of the results section has a breakdown of the diagnoses and a discussion regarding how these concepts are represented and their meaning for caregivers of each diagnoses. Please see p.35-39

Are the discussion and conclusions justified by the results Page 29- "All caregivers described not knowing who to contact"- appears inaccurate to say this?- This sentence has been removed

The first para of the discussion appears to be a repetition of the results. Greater interpretation is needed and it may be more effective to integrate some of the references included on page 30. – The discussion section has been re-edited, please see p. 39-41

All of the data provided from para 2 of page 29 is new and should be presented as part of the results rather than discussion. – This has been amended as per your suggestion. Please see p. 39-41

Why would it be suggested that guidelines be developed for engaging with caregivers of CAD patients only? – This is point has been clarified, please see p. 39 under CAD and p.41, paragraph one under implications for practice

Unclear what is meant by "Qualitative research clearly demonstrates the caregiving experience is constantly evolving and this needs to be reflected in quantitative research"- This sentence has been amended, please see p. 41, paragraph 2

Are the study limitations discussed adequately?

The final point of the study limitations section is more a limitation of the included studies rather than a limitation of the review. A separate section dedicated to the methodological quality and limitations of included studies is warranted. – This section has now been further edited to clarify study and review limitation. Please see p. 40 & 41, paragraph 3 &4 under sub-heading strengths and limitations

Minor points

In the Abstract, the eligibility criteria for selecting studies- should this be "providing care for patients with HF, COPD or CAD"? Thank you, this has been changed to "or" throughout the text.

First sentence of introduction is awkward to read- please consider revising. – This has been rewritten, please see p.4 paragraph 1.

"The Institute of Public Care examining a Caregivers' Life (2017)"quote in introduction- unclear what this document is. It also needs to be included in the list of references. – This is clarified in p.4 paragraph 2 and is included in the reference list.

It would be helpful to have more explanation of the "transitions and the dynamic process of caring" which you mention in the introduction. – This has been elaborated upon. Please see p. 4 paragraph 2.

The web addresses provided in the Methods section should instead be provided in the reference list. – This has been amended

In Study quality assessment- "This contributes to a synthesis of evidence which will facilitate obtaining a summary of evidence."- these two things mean that same thing? – This sentence has been edited to clarify this point. Please see p. 6, paragraph 3

Typo in Table 1- "Holistic" - Corrected

Some inconsistent use of caregiver/carer throughout – This has been corrected. The term caregiver is now used throughout the paper

Overall more editorial work and proofing is needed.

Reviewer: 2

Reviewer Name: Florence Potier

Institution and Country: CHU UCL Namur, Institute of Health and Society, Belgium
Competing Interests: None declared

Dear Ms. Potier

Many thanks for reviewing this paper and for your comments. I have made amendments as per your suggestions.

The explores a very interesting topic.

The study methods are appropriate.

However, I have some suggestions to improve the clarity of the presentation.

Results:

1. Table 2: could you add information's about the mean age of the care-receiver and how long the caregiving has occurred? These patient data help clinicians in interpreting the results or implement them in clinical practice. – This has been amended, please see Table 2 p. 12-24
2. Table 3 : could you add the different designs (to state this important limitation) – This has been amended as per your suggestion. Please see Table 3, p. 26
3. Table 3: The classical presentation is “CAD, n (%) 6 (11.1)” - This has been amended as per your suggestion. Please see Table 3, p. 26
4. Figure 2 can be improved (groups with intersections for example. The arrow between “qual and quant papers” and “qual papers only” is not clear.)- Figure 2 has been re-edited to clarify the associations. Please see p. 33

Discussion:

1. The summary of the findings by diagnosis should be placed in the results section (or in any case before the first sentence: “this mixed methods...condition”.) – This has been moved into the results section as per your suggestion. Please see p. 38/39
2. Please, add some “Implications for practice” in the section “implication for practice an future research” (new ideas should not be only in conclusions) – The implications for practice paragraph has been elaborated, please see pages 41.

I hope the authors will find these comments useful

VERSION 2 – REVIEW

REVIEWER	Florence Potier Institute of Health and Society, Brussels, Belgium
REVIEW RETURNED	16-Feb-2018

GENERAL COMMENTS	This manuscript has improved substantially. However, there are still little problems in presentation of the Table 3  - No decimal for the N - The same number of decimals for % - No repetition of "%" after the number (already specified at the beginning of the line)
---

REVIEWER	Elise Mansfield University of Newcastle, Australia
REVIEW RETURNED	09-Mar-2018

GENERAL COMMENTS	Major points Are methods described sufficiently? I still do not feel it is accurate to describe this as a review of only predictors of caregiver experience. While understanding the factors associated with greater burden is important, it is also pertinent to examine the overall type and extent of burdens which are identified. Supporting this point, not all studies reported in Table 2 are described as 'correlational'- some simply describe the prevalence of different burdens and other experiences. Is study design appropriate? I was still not clear on a number of aspects of the quality assessment of quantitative studies:  - it is unclear exactly what the quality criteria were and the minimum standard they had to meet in order to receive a score of 1. For example how was an 'appropriate sampling strategy' defined? - There appears to be one criteria (study design?) missing from the list described in the Method. - one criteria is 'level of attrition', however based on checking a couple of papers I think the more appropriate term for this would be 'response rate' or 'consent rate'. Attrition usually refers to loss of participants over time during a study, such as in longitudinal studies when participants are lost to follow up. Without clarification of the above it is not possible to determine whether there has been a fair assessment of study quality. In their response the authors indicate that no low quality studies were included in the narrative synthesis. However Table 2b appears to show that low quality studies were included. If some studies are indeed of sub-par quality, then a case could be made for excluding these studies from Table 2, and removing these from the narrative analysis. This would ensure that the results of the review are based only on the highest quality studies. I am not convinced about how the narrative formation technique was used to generate concepts for quantitative studies. In the example provided in Table 1, the concept generated from the 'modal' profile which describes the group being studied is 'role'. However given that
--

all included studies include caregivers as the participant group, wouldn't this mean that all would be coded as 'role'? I would have thought this theme would be more related to caregivers' perception of their role and the tasks involved? Also it is not clear how the association between patient depression and caregiver burden can be used to generate the concept 'relationships'. I am still not convinced that this technique provides an accurate portrayal of the concepts included in quantitative studies.

Do results address the research question/presented clearly?

Table 2- the authors have reported key conclusions from each of the papers. This appears to present a combination of key results and the authors' subjective conclusions based on the results. Reporting on the authors' subjective conclusions may introduce bias to the results. It would present a more objective picture if only the key findings from included studies were presented (similar to the way these are presented in row 2 of Table 1).

Given the scope of studies included in the review, a greater attempt to summarise the findings under each of the themes appears warranted. In particular it would be interesting to include more prevalence estimates from the quantitative data to get more of a sense of a severity of each issue, as well as the factors associated with higher levels of difficulties experienced. Based on the presentation of results we only know the number of studies in which an issue was reported, not how common it was among the participants surveyed.

References appropriate?

The breakdown of the results by disease type is informative, however this section requires references.

Limitations discussed adequately?

Under strengths and limitations, would find it helpful to perhaps separate into limitations of the included studies vs limitations of the review. These are quite different things. In particular, additional commentary on the level of quality of the included studies, and areas in which study quality was generally poor would be informative for informing future research in this area.

It is unclear how exclusion of low quality studies would be a limitation of the review.

Discussion/conclusions justified?

The second part of the first paragraph of 'Implications for practice and future research' appears very generic. Greater specificity is needed - for example it appears that quite a lot of research has already been done looking at mental health. Greater synthesis of the results under each theme would assist in showing the specific gaps in research which are deserving of further attention.

Additional discussion on how these results could be used to formulate helpful interventions targeting caregivers seems warranted.

Minor points

It is still unclear about whether assessment of articles against the eligibility criteria was cross checked by another person. Currently it still appears that the screening was only conducted by one author.

	Table 1- the explanation of the 'normative' profile from the analysis framework suggests that it should be looking at how the individuals in the study compare to the general population (e.g. do they show higher levels of burden than people with similar age/gender who are not caregivers?). This is not reflected in the example data extraction provided. Labeling of this table could also be more descriptive. In the PRISMA diagram it is unclear which is meant by the exclusion criterion 'Do not meet critical appraisal criteria'
--	--

VERSION 2 – AUTHOR RESPONSE

Manuscript ID bmjopen-2017-020927.R1 "Who Cares?" The Experiences of Caregivers of Adults Living with Heart Failure, Chronic Obstructive Pulmonary Disease and Coronary Artery Disease – A Mixed Methods Systematic Review"

Author's response to reviewer comments - 20th March 2018

We thank the two peer reviewers for their additional comments. We have carefully reviewed these latest comments and provide a point-by-point reply below. We also attach a tracked edited version of the manuscript with tracked comprehensive edits in order take account of these additional comments.

Reviewer 1

Major points

Are methods described sufficiently?

I still do not feel it is accurate to describe this as a review of only predictors of caregiver experience. While understanding the factors associated with greater burden is important, it is also pertinent to examine the overall type and extent of burdens which are identified. Supporting this point, not all studies reported in Table 2 are described as 'correlational'- some simply describe the prevalence of different burdens and other experiences. We apologise that we have not have been sufficiently accurate clear in our description of our study. We have rephrased the description of the study in the strengths and limitations box (p.3) as a review of the type and extent of caregiver's experiences and also of predictors of caregiver's experience.

Is study design appropriate?

I was still not clear on a number of aspects of the quality assessment of quantitative studies:

- it is unclear exactly what the quality criteria were and the minimum standard they had to meet in order to receive a score of 1. For example how was an 'appropriate sampling strategy' defined? This has been amended to provides clarity on scoring see pg. 6 under sub heading "study quality assessment"

- There appears to be one criteria (study design?) missing from the list described in the Method. As above

- one criteria is 'level of attrition', however based on checking a couple of papers I think the more appropriate term for this would be 'response rate' or 'consent rate'. Attrition usually refers to loss of participants over time during a study, such as in longitudinal studies when participants are lost to follow up. As above

Without clarification of the above it is not possible to determine whether there has been a fair assessment of study quality. As above.

In their response the authors indicate that no low quality studies were included in the narrative synthesis. However Table 2b appears to show that low quality studies were included. If some studies are indeed of sub-par quality, then a case could be made for excluding these studies from Table 2, and removing these from the narrative analysis. This would ensure that the results of the review are based only on the highest quality studies. We would like to clarify that only high quality qualitative studies were included, whilst for quantitative studies we included studies of all quality. The rationale for this was a pragmatic one - the volume of qualitative studies judged to be of high quality was relatively high (i.e. 21 studies) and in accord with current guidelines for the synthesis of qualitative evidence, we limited inclusion of qualitative studies to high quality. In contrast, only 4 quantitative studies were judged to be of high quality, and to ensure comprehensiveness of our review, we decided to include all quantitative studies regardless of quality. We have now clarified this rationale in the quality assessment text in the results section. Given the reviewer's comment, we also now note in the discussion section that the conclusions from our review would have remained consistent if we had limited our synthesis to only high quality quantitative studies.

I am not convinced about how the narrative formation technique was used to generate concepts for quantitative studies. In the example provided in Table 1, the concept generated from the 'modal' profile which describes the group being studied is 'role'. However given that all included studies include caregivers as the participant group, wouldn't this mean that all would be coded as 'role'? I would have thought this theme would be more related to caregivers' perception of their role and the tasks involved? Also it is not clear how the association between patient depression and caregiver burden can be used to generate the concept 'relationships'. I am still not convinced that this technique provides an accurate portrayal of the concepts included in quantitative studies.

We apologise that there remains a lack of clarity on the use of the narrative formation approach. A narrative formation approach was utilised to convert the quantitative data to qualitative due to the heterogeneity of the quantitative studies included in this review, therefore it was not feasible to produce a meta-analysis. Reading and re-reading quantitative papers enabled the authors to identify narrative elements of these studies and rigorously extract what was presented. To reflect this, additional information has been added to Table 1, pgs.7-10.

Do results address the research question/presented clearly?

Table 2- the authors have reported key conclusions from each of the papers. This appears to present a combination of key results and the authors' subjective conclusions based on the results. Reporting on the authors' subjective conclusions may introduce bias to the results. It would present a more objective picture if only the key findings from included studies were presented (similar to the way these are presented in row 2 of Table 1). We agree with the reviewer - we retained the aim of each study (as stated by study authors) and deleted the key findings column of Table 2, pgs. 13-24

Given the scope of studies included in the review, a greater attempt to summarise the findings under each of the themes appears warranted. In particular it would be interesting to include more prevalence estimates from the quantitative data to get more of a sense of a severity of each issue, as well as the factors associated with higher levels of difficulties experienced. Based on the presentation of results we only know the number of studies in which an issue was reported, not how common it was among the participants surveyed. We have carefully reflected on this comment by the reviewer. As acknowledged, we do currently provide the prevalence of quantitative and qualitative studies under

each theme. However, given the heterogeneity of reporting of quantitative studies, not only would a more granular estimate of prevalence be technically very difficult to operationalise but we believe would also impact the scope of our current mixed methods synthesis. We would note previous published systematic reviews that have followed similar synthesis approach to our study:

Grant, J., Graven L. (2018) Problems experienced by informal caregivers of individuals with heart failure: An integrative review. *International journal of nursing studies*. 41-66.
<https://doi.org/10.1016/j.ijnurstu.2017.12.016>

Karimi, M. & Clark, A.M. (2016) How do patients' values influence heart failure self-care decision-making?: A mixed-methods systematic review. *International journal of nursing studies*. 89-104.
<http://dx.doi.org/10.1016/j.ijnurstu.2016.03.0100>

References appropriate?

The breakdown of the results by disease type is informative, however this section requires references. We have amended this section. Please see pgs. 38-39 sub-headings "Heart Failure", "Chronic Obstructive Pulmonary Disease" and "Coronary Artery Disease".

Limitations discussed adequately?

Under strengths and limitations, would find it helpful to perhaps separate into limitations of the included studies vs limitations of the review. These are quite different things. In particular, additional commentary on the level of quality of the included studies, and areas in which study quality was generally poor would be informative for informing future research in this area. We agree with the reviewer and have amended this section, see pg. 40-41

It is unclear how exclusion of low quality studies would be a limitation of the review. – This sentence has been removed and discussion of the limitations of study quality redrafted, please see pg. 40-41

Discussion/conclusions justified?

The second part of the first paragraph of 'Implications for practice and future research' appears very generic. Greater specificity is needed - for example it appears that quite a lot of research has already been done looking at mental health. Greater synthesis of the results under each theme would assist in showing the specific gaps in research which are deserving of further attention.

Additional discussion on how these results could be used to formulate helpful interventions targeting caregivers seems warranted. We agree with the reviewer and amended the "implications for practice and future research" paragraph. P.41

Minor points

It is still unclear about whether assessment of articles against the eligibility criteria was cross checked by another person. Currently it still appears that the screening was only conducted by one author. The reviewer is correct that the initial screening was undertaken by a single reviewer. However, only studies that very obviously did not meet the inclusion/exclusion criteria were excluded at this point.

Full papers were sought for all potential included titles and abstracts and reviewed by two reviewers. We have clarified this on p. 5 under sub heading “search strategy”

Table 1- the explanation of the 'normative' profile from the analysis framework suggests that it should be looking at how the individuals in the study compare to the general population (e.g. do they show higher levels of burden than people with similar age/gender who are not caregivers?). This is not reflected in the example data extraction provided. We agree with the reviewer and we have added further clarification has been added to normative profile in Table 1, pgs. 7-10.

Labeling of this table could also be more descriptive. We have added further clarification has been added to normative profile in Table 1, pg. 7-10.

In the PRISMA diagram it is unclear which is meant by the exclusion criterion 'Do not meet critical appraisal criteria' We agree with the reviewer and have clarified the PRISMA diagram

Reviewer: 2

Reviewer Name: Florence Potier

This manuscript has improved substantially. Thank you.

However, there are still little problems in presentation of the Table 3

- No decimal for the N
- The same number of decimals for %
- No repetition of "%" after the number (already specified at the beginning of the line)

We agree. Table 3 has now been amended in accord with the reviewer’s suggestions, pgs. 26-27

VERSION 3 – REVIEW

REVIEWER	Elise Mansfield University of Newcastle, Australia
REVIEW RETURNED	12-May-2018

GENERAL COMMENTS	Thank you to the authors for thoughtfully responding to my comments. I had a few further comments: Methods described sufficiently? I am still not entirely clear on how the narrative formation technique was applied- e.g. the 'caregiver role' concept is later described in the results as findings related to role change/loss, and the tasks involved. It is not clear how the example study provided examines this concept, as it appears to be a study about the relationship between patient depression and caregiver burden. In addition I am unclear as to why the finding that caregivers of depressed patients reported greater burden is reported across three of the profiles- wouldn't this only fit with the 'comparative' profile? Without a clear understanding of how the concepts were generated from the quantitative studies it is difficult to trust the accuracy of concepts identified for each study. Perhaps it would help to have more description about how the
--

	concepts are derived from each of the profiles (i.e. the steps to go through to generate the concepts for each profile). A minor point- my understanding is that Table 1 presents an example of the approach applied to one study. If this is correct, I think it is potentially confusing to have a column dedicated to the study reference- perhaps this could just be in the table caption? Results presented clearly? Figures 1 and 2 did not appear in the latest version of the manuscript and so I could not check any revisions to these. Re Table 2, perhaps the authors misunderstood my suggestion, which was to reformat the study findings so that they present only the data produced by the study, rather than the authors' subjective interpretation of findings. Given only a brief summary of the findings can be presented in the Results text section, I believe it is still important to present the individual study data somewhere for transparency, and for interested readers. It may also be helpful to include a column to identify the concepts generated from each study. Minor point- the first paragraph of the figure legend for Table 2 does not appear to map to this Table. Discussion/Conclusions I still believe it would be appropriate to include greater specificity in the Implications section, to give researchers and providers some suggestions for how they could use the results to develop interventions- e.g. which concepts most frequently emerged and the possible types of interventions that might be helpful in addressing these concepts. It may also be useful to talk about differences in the concepts that emerged for qualitative vs quantitative studies and what this means for future studies in the field. Page 42- The following two sentences appear to say the same thing: "Future research should focus on the involvement of caregivers in the design and delivery of interventions for patients with cardiopulmonary disease. In addition to targeting patients, interventions need to consider caregivers. " Study limitations It is unclear how inclusion of the lower quality studies would increase generalisability of the review? Publication ethics Please ensure that direct quotes reported in Table 1 are appropriately referenced.
--	--

VERSION 3 – AUTHOR RESPONSE

Dear reviewer

Many thanks for reviewing this paper and for your comments.

Methods described sufficiently?

I am still not entirely clear on how the narrative formation technique was applied- e.g. the 'caregiver role' concept is later described in the results as findings related to role change/loss, and the tasks involved. It is not clear how the example study provided examines this concept, as it appears to be a study about the relationship between patient depression and caregiver burden. In addition I am unclear as to why the finding that caregivers of depressed patients reported greater burden is reported across three of the profiles- wouldn't this only fit with the 'comparative' profile? Without a clear understanding of how the concepts were generated from the quantitative studies it is difficult to trust the accuracy of concepts identified for each study - Table 1 has been edited to make this clearer. Additionally in supplementary file 1 tables 2a,b, and c map which study generated which concept.

A minor point- my understanding is that Table 1 presents an example of the approach applied to one study. If this is correct, I think it is potentially confusing to have a column dedicated to the study reference- perhaps this could just be in the table caption? – This has been amended and now inserted as: Narrative Formation (Example: Chung, 2016)

Figures 1 and 2 did not appear in the latest version of the manuscript and so I could not check any revisions to these. – These were uploaded to scholar one author centre.

Re Table 2, perhaps the authors misunderstood my suggestion, which was to reformat the study findings so that they present only the data produced by the study, rather than the authors' subjective interpretation of findings. Given only a brief summary of the findings can be presented in the Results text section, I believe it is still important to present the individual study data somewhere for transparency, and for interested readers. It may also be helpful to include a column to identify the concepts generated from each study. – A new supplementary file 2 (table 3) has been inserted to address individual study results.. Concepts table included as supplementary file 2a,b and c

Minor point- the first paragraph of the figure legend for Table 2 does not appear to map to this Table.- Amended

I still believe it would be appropriate to include greater specificity in the Implications section, to give researchers and providers some suggestions for how they could use the results to develop interventions- e.g. which concepts most frequently emerged and the possible types of interventions that might be helpful in addressing these concepts. - Amended p.40, paragraph 2

It may also be useful to talk about differences in the concepts that emerged for qualitative vs quantitative studies and what this means for future studies in the field.-Amended p.41

The following two sentences appear to say the same thing: "Future research should focus on the involvement of caregivers in the design and delivery of interventions for patients with cardiopulmonary disease. In addition to targeting patients, interventions need to consider caregivers." – Amended. Second sentence has been removed. p.40

It is unclear how inclusion of the lower quality studies would increase generalisability of the review? Sentence amended, paragraph 1, pg. 40.

Please ensure that direct quotes reported in Table 1 are appropriately referenced. - Amended